# Genomic Alterations and Microbiota Crosstalk in Hepatic Cancers: The Gut–Liver Axis in Tumorigenesis and Therapy

**DOI:** 10.3390/genes16080920

**Published:** 2025-07-30

**Authors:** Yuanji Fu, Jenny Bonifacio-Mundaca, Christophe Desterke, Íñigo Casafont, Jorge Mata-Garrido

**Affiliations:** 1Institut Necker Enfants Malades, INSERM, CNRS, Université Paris Cité, F-75015 Paris, France; yuanji.fu@inserm.fr; 2National Tumor Bank, Department of Pathology, National Institute of Neoplastic Diseases, Surquillo 15038, Peru; jenny.bonifacio@upch.pe; 3Faculté de Médecine du Kremlin Bicêtre, University Paris-Sud, Université Paris-Saclay, 94270 Le Kremlin-Bicêtre, France; christophe.desterke@inserm.fr; 4Cell and Tissue Biology Group, Anatomy and Cell Biology Department, University of Cantabria-IDIVAL, 39011 Santander, Spain; inigo.casafont@unican.es

**Keywords:** hepatocellular carcinoma, intrahepatic cholangiocarcinoma, cancer, microbiota, gut–liver axis

## Abstract

**Background/Objectives:** Hepatic cancers, including hepatocellular carcinoma (HCC) and cholangiocarcinoma (CCA), are major global health concerns due to rising incidence and limited therapeutic success. While traditional risk factors include chronic liver disease and environmental exposures, recent evidence underscores the significance of genetic alterations and gut microbiota in liver cancer development and progression. This review aims to integrate emerging knowledge on the interplay between host genomic changes and gut microbial dynamics in the pathogenesis and treatment of hepatic cancers. **Methods:** We conducted a comprehensive review of current literature on genetic and epigenetic drivers of HCC and CCA, focusing on commonly mutated genes such as *TP53*, *CTNNB1*, *TERT*, *IDH1/2*, and *FGFR2*. In parallel, we evaluated studies addressing the gut–liver axis, including the roles of dysbiosis, microbial metabolites, and immune modulation. Key clinical and preclinical findings were synthesized to explore how host–microbe interactions influence tumorigenesis and therapeutic response. **Results:** HCC and CCA exhibit distinct but overlapping genomic landscapes marked by recurrent mutations and epigenetic reprogramming. Alterations in the gut microbiota contribute to hepatic inflammation, genomic instability, and immune evasion, potentially enhancing oncogenic signaling pathways. Furthermore, microbiota composition appears to affect responses to immune checkpoint inhibitors. Emerging therapeutic strategies such as probiotics, fecal microbiota transplantation, and precision oncology based on mutational profiling demonstrate potential for personalized interventions. **Conclusions:** The integration of host genomics with microbial ecology provides a promising paradigm for advancing diagnostics and therapies in liver cancer. Targeting the gut–liver axis may complement genome-informed strategies to improve outcomes for patients with HCC and CCA.

## 1. Introduction

Hepatic cancers are among the most aggressive and fatal malignancies worldwide, with hepatocellular carcinoma (HCC) and cholangiocarcinoma (CCA) constituting the most prevalent primary liver tumors [1,2,3]. Together, these malignancies are associated with high mortality rates due to late-stage presentation, limited treatment options, and frequent recurrence [1,3]. The liver, a vital organ responsible for numerous metabolic, detoxifying, and synthetic functions, is particularly vulnerable to various pathological conditions that can drive malignant transformation. Among primary hepatic cancers, HCC accounts for 75–85% of cases, while CCA, which arises from the bile ducts, constitutes 10–15% [2]. These two entities differ in their cellular origins and underlying risk factors, but both share common themes of chronic inflammation, fibrosis, and genetic instability.

HCC originates from hepatocytes, the main parenchymal cells of the liver, and is strongly associated with chronic liver disease, particularly cirrhosis [4,5]. Key risk factors for HCC include chronic infection with the hepatitis B virus (HBV) and hepatitis C virus (HCV), excessive alcohol consumption, and the growing prevalence of non-alcoholic fatty liver disease (NAFLD) [2,6,7]. While HBV remains the predominant cause of HCC in East Asia and sub-Saharan Africa, HCV, alcohol, and NAFLD are more prominent contributors in Western populations [2]. Regardless of etiology, cirrhosis represents the most significant predictor of HCC development, with an estimated annual incidence of 1–5% among cirrhotic patients [3,4]. The World Health Organization (WHO) reports over 900,000 new cases of HCC and nearly 830,000 deaths annually, underscoring its significant global health impact [3].

CCA arises from cholangiocytes lining the intrahepatic and extrahepatic bile ducts and can be further classified anatomically into intrahepatic (iCCA), perihilar (pCCA), and distal (dCCA) subtypes depending on tumor location [8,9,10]. Although less common than HCC, CCA carries an equally poor prognosis due to its typically insidious onset, late diagnosis, and resistance to conventional therapies [8]. Incidence trends for CCA show regional and subtype-specific variation: iCCA has increased globally over recent decades, while extrahepatic forms (pCCA and dCCA) have remained stable or declined [9]. CCA is strongly associated with chronic biliary inflammation, including conditions such as primary sclerosing cholangitis (PSC), hepatolithiasis, and liver fluke infections (e.g., Opisthorchis viverrini and Clonorchis sinensis) in endemic areas of Southeast Asia [8,9,10]. Additional risk factors include chronic liver diseases such as HBV, HCV, and NAFLD, as well as exposure to carcinogens like thorotrast, asbestos, and nitrosamines [11,12,13,14,15,16].

Despite distinct etiologies and tissue origins, both HCC and CCA share common pathophysiological mechanisms. Chronic liver injury triggers persistent inflammation, activation of hepatic stellate cells, and extracellular matrix remodeling, leading to fibrosis and ultimately cirrhosis [17,18]. This environment fosters genetic and epigenetic alterations that promote malignant transformation. Moreover, cycles of hepatocellular death and regeneration increase the likelihood of acquiring mutations in key oncogenic pathways [19].

In this context, the gut–liver axis has emerged as a crucial contributor to liver disease progression and hepatic tumorigenesis. The liver is anatomically and functionally linked to the intestine via the portal vein, receiving not only nutrients and metabolites but also microbial products such as lipopolysaccharides (LPS); microbial DNA; and various metabolites, including short-chain fatty acids (SCFAs), secondary bile acids, and ammonia [20,21,22]. This continuous exposure allows for direct microbial influence on hepatic immune responses, metabolic processes, and inflammation. Dysbiosis—disruption of the normal gut microbiota—can enhance microbial translocation and stimulate hepatic inflammation, thereby contributing to fibrosis and cancer development [14,16,23].

Recent studies have revealed that alterations in the gut microbiota may affect not only liver injury and fibrosis but also specific oncogenic signaling pathways, immune surveillance, and therapeutic responsiveness in hepatic cancers. These insights have led to growing interest in the gut–liver axis as a therapeutic target. Interventions such as probiotics, prebiotics, synbiotics, antibiotics, and fecal microbiota transplantation (FMT) are currently being investigated for their potential to modify disease progression and improve treatment outcomes in liver cancer patients.

The aim of this review is to provide a comprehensive analysis of the interplay between genomic alterations and gut microbiota in hepatic cancers, with a focus on HCC and CCA. We begin by summarizing the current understanding of the gut–liver axis and its role in liver health. We then explore how dysbiosis and microbial metabolites contribute to chronic liver disease and carcinogenesis. In addition, we evaluate current and emerging strategies that target the microbiota for therapeutic benefit and propose future research directions to advance this promising field. By consolidating recent findings, this review seeks to enhance our understanding of the gut–liver axis in hepatic oncology and guide future clinical and translational efforts.

## 2. Materials and Methods

This review was conducted in accordance with established guidelines for narrative reviews and aimed to provide a comprehensive synthesis of current knowledge on the role of gut microbiota in hepatic cancer development and therapy. The methodology followed is outlined below.

### 2.1. Literature Search Strategy

A systematic literature search was performed across multiple scientific databases, including PubMed, Web of Science, Scopus, and Google Scholar, for relevant peer-reviewed articles published up to March 2024. The primary keywords and search terms included combinations of “hepatic cancer”, “hepatocellular carcinoma”, “cholangiocarcinoma”, “gut microbiota”, “gut-liver axis”, “dysbiosis”, “microbial metabolites”, “bile acids”, “inflammation”, “immune modulation”, “probiotics”, “prebiotics”, “synbiotics”, “fecal microbiota transplantation”, and “immune checkpoint inhibitors”. Boolean operators such as AND, OR, and NOT were used to refine the searches.

### 2.2. Inclusion and Exclusion Criteria

Articles were included based on the following criteria:Published in English in peer-reviewed journals.Focused on hepatocellular carcinoma (HCC), cholangiocarcinoma (CCA), or general hepatic cancers.Addressed the relationship between gut microbiota and hepatic cancer pathogenesis, diagnosis, prognosis, or treatment.Included original research studies, clinical trials, systematic reviews, and meta-analyses.

Excluded studies included:Non-peer-reviewed materials (e.g., conference abstracts, editorials, or letters).Studies focusing on unrelated gastrointestinal malignancies or non-microbial factors without clear relevance to the gut–liver axis.Animal studies lacking translational relevance or mechanistic insight applicable to humans.

### 2.3. Data Extraction and Synthesis

Relevant data from selected studies were extracted manually and organized thematically into conceptual domains: (1) hepatic cancer pathogenesis and risk factors, (2) dysbiosis and the gut–liver axis, (3) microbial metabolites and immune modulation, (4) diagnostic and prognostic applications of microbiota profiling, and (5) microbiota-targeted therapies. Particular attention was given to recent findings on microbial signatures associated with HCC and CCA and their potential integration into clinical practice.

To ensure the scientific rigor and comprehensiveness of the review, the included references were cross-verified, and overlapping themes were synthesized into a coherent narrative structured around the evolving understanding of the microbiota’s role in hepatic oncology.

### 2.4. Limitations

As a narrative review, this article does not include a quantitative meta-analysis or systematic risk of bias assessment. Nevertheless, every effort was made to incorporate the most recent and impactful findings from high-quality studies to ensure an accurate and representative overview of the field.

## 3. Results

### 3.1. Hepatic Cancer Pathogenesis

#### 3.1.1. Hepatocellular Carcinoma

The incidence of HCC varies significantly depending on geographical regions, closely linked to differences in chronic liver disease prevalence. Chronic viral hepatitis, especially caused by HBV and HCV, is among the most significant risk factors for HCC. HBV, particularly in regions like Asia and Africa, is a key driver of HCC due to its direct oncogenic potential [24,25]. HBV integrates into the host’s genome, causing genomic instability that can lead to mutations in important regulatory genes, such as tumor suppressor genes. Chronic HBV infection also triggers continuous liver inflammation, which promotes immune-mediated liver damage, fibrosis, and cirrhosis, creating a favorable environment for cancer development [24,25].

HCV plays a similar role, albeit through different mechanisms. Chronic HCV infection results in prolonged liver inflammation and fibrosis, culminating in cirrhosis, a well-established risk factor for HCC [26,27]. Unlike HBV, HCV does not integrate into the host’s DNA but induces oxidative stress, mitochondrial dysfunction, and endoplasmic reticulum stress, contributing to DNA damage and cancer development [26,27]. The annual risk of HCC in cirrhotic patients with chronic HCV infection is notably high, estimated to range between 1 and 5% [26,27].

Alcohol consumption is another major contributor to HCC risk [28]. Chronic alcohol abuse leads to liver damage that progresses from steatosis (fatty liver) to alcoholic steatohepatitis (ASH), fibrosis, and ultimately cirrhosis [29]. Fibrosis and cirrhosis are the most critical risk factors, with patients facing an annual HCC development risk of 3–10% [30]. Fibrosis results from excessive accumulation of extracellular matrix (ECM) proteins following chronic liver injury, primarily driven by the activation of hepatic stellate cells (HSCs) [31]. Over time, fibrosis progresses to cirrhosis, characterized by distorted liver architecture [30]. In cirrhotic livers, increased activity of fibrogenic growth factors like TGF-β and angiogenic factors such as VEGF promotes tumor growth and neovascularization, facilitating cancer progression [30,31].

Alcohol promotes liver cancer through multiple mechanisms, including inducing chronic inflammation, oxidative stress, and direct toxicity to liver cells [29]. In particular, alcohol metabolism results in the production of acetaldehyde, a toxic compound that forms DNA adducts, potentially leading to mutations that drive carcinogenesis [29].

NAFLD, and its more severe form, non-alcoholic steatohepatitis (NASH), have emerged as significant risk factors for HCC, especially in the context of the global obesity epidemic [32,33]. NAFLD is closely associated with metabolic syndromes like obesity, type 2 diabetes, and dyslipidemia [32]. This disease spectrum ranges from simple steatosis to more advanced NASH, which can progress to fibrosis, cirrhosis, and eventually HCC. While cirrhosis is a significant risk factor, it is noteworthy that HCC can develop in NAFLD patients, even in the absence of cirrhosis [34]. Mechanisms involved in NAFLD-related HCC include lipotoxicity, insulin resistance, oxidative stress, and chronic inflammation [35]. Free fatty acids and other harmful molecules accumulate in hepatocytes, leading to mitochondrial dysfunction and triggering cell death pathways, such as apoptosis and necrosis, further contributing to cancer development [36].

Other notable risk factors include cirrhosis and aflatoxin exposure, particularly aflatoxin B1, produced by Aspergillus species in contaminated foods like grains and nuts, which is another potent carcinogen [37]. Exposure to aflatoxin significantly raises HCC risk, especially when combined with chronic HBV infection [38]. Aflatoxin contributes to carcinogenesis through mutations in the TP53 tumor suppressor gene [38].

HCC development is further influenced by genetic and epigenetic alterations. Somatic mutations in critical genes regulating cell growth, survival, and differentiation are common in HCC [19,39]. For example, mutations in TP53 result in the loss of the gene’s ability to regulate the cell cycle and maintain genomic stability, allowing uncontrolled hepatocyte proliferation [40]. Mutations in the CTNNB1 gene activate the Wnt/β-catenin signaling pathway, which promotes cancer cell growth and inhibits apoptosis [40,41]. Additionally, mutations in TERT, the gene encoding telomerase reverse transcriptase, are found in a significant percentage of HCC cases, facilitating telomere maintenance and allowing cancer cells to evade replicative senescence [33,42]. In addition to genetic mutations, epigenetic modifications, such as DNA methylation and histone acetylation, play a role in HCC development. Hypermethylation of tumor suppressor genes like RASSF1A silences critical pathways involved in controlling the cell cycle and apoptosis, promoting cancerous growth [42]. Furthermore, oxidative stress caused by chronic liver injury and metabolic disturbances leads to the excessive production of ROS, which further damages cellular DNA, lipids, and proteins. This oxidative stress is particularly evident in viral hepatitis and NAFLD, where chronic inflammation and metabolic dysfunction generate ROS that contribute to DNA damage and activate oncogenic pathways [43,44].

In summary, the pathogenesis of hepatocellular carcinoma is a complex interplay of chronic inflammation, fibrosis, cirrhosis, and genetic alterations (Figure 1). Risk factors such as chronic viral infections (HBV and HCV), alcohol consumption, and NAFLD create a pro-carcinogenic environment through mechanisms of DNA damage, immune dysregulation, and excessive cellular proliferation.

##### Genetic and Genomic Alterations in Hepatocellular Carcinoma

HCC exhibits a complex genomic landscape with numerous recurrent genetic mutations and chromosomal alterations that drive liver oncogenesis. The most frequently mutated gene in HCC is *TP53*, present in approximately 30–50% of cases. TP53 mutations disrupt its role as a tumor suppressor, impairing cell cycle arrest and apoptosis in response to DNA damage [45].

Another major genetic alteration involves *CTNNB1*, which encodes β-catenin and activates the Wnt/β-catenin signaling pathway, promoting hepatocyte proliferation and resistance to apoptosis [46]. Mutations in *TERT* promoter regions (telomerase reverse transcriptase) are among the earliest events in hepatocarcinogenesis and lead to telomerase reactivation, supporting unlimited replicative potential [47].

Other commonly altered genes include *AXIN1*, *ARID1A*, *KEAP1*, and *NFE2L2*, which are involved in chromatin remodeling, oxidative stress response, and epigenetic regulation [48]. High-throughput sequencing has also revealed copy number variations (amplification of *MYC* and deletions in *CDKN2A*) and mutational signatures linked to aflatoxin exposure and chronic hepatitis B infection [49].

In HBV-related HCC, the integration of viral DNA into the host genome induces genomic instability, promotes insertional mutagenesis, and affects nearby oncogenes or tumor suppressors. For example, HBV integration near *TERT*, *MLL4*, or *CCNE1* can drive hepatocarcinogenesis.

Epigenetic changes are also prominent in HCC, with DNA hypermethylation silencing key tumor suppressor genes (e.g., *RASSF1A* and *CDKN2A*) and histone modifications altering gene expression patterns [50]. The interplay between chronic liver injury, inflammation, and DNA damage promotes the accumulation of these mutations, laying the foundation for malignant transformation.

#### 3.1.2. Cholangiocarcinoma

CCA is a highly aggressive cancer that arises from the epithelial cells lining the bile ducts [2,8]. It represents approximately 10–15% of all primary liver cancers and is the second most common liver malignancy after HCC [51]. CCA is characterized by its poor prognosis due to the typically late stage at diagnosis, rapid disease progression, and limited treatment options. The median survival for patients with CCA is often less than two years, and the 5-year survival rate remains dismal, typically below 10% [52]. There are two primary subtypes of CCA: iCCA, which occurs within the bile ducts located inside the liver [9], and eCCA [10], which occurs in the bile ducts outside the liver. Both subtypes share common pathogenic mechanisms but differ in their anatomical location, clinical presentation, and treatment strategies.

Risk factors for CCA vary depending on geographical regions, but common conditions linked to the development of CCA include primary sclerosing cholangitis (PSC), chronic liver fluke infections, and cirrhosis [51]. PSC, a chronic inflammatory disease that leads to scarring and stricture formation in the bile ducts, is one of the most well-established risk factors, particularly in Western countries [53]. PSC is often associated with inflammatory bowel disease (IBD), especially ulcerative colitis, and patients with PSC have a markedly increased risk of developing CCA, with estimates suggesting that 10–15% of PSC patients will eventually develop this malignancy [54,55]. In Southeast Asia, chronic infection with liver flukes such as Opisthorchis viverrini and Clonorchis sinensis is a significant risk factor for CCA [56]. These parasites are acquired through the consumption of raw or undercooked freshwater fish, and long-term infestation leads to chronic bile duct inflammation, hyperplasia, and an increased risk of carcinogenesis [56]. In endemic areas such as Thailand, liver fluke infections are a leading cause of CCA, and preventive measures aimed at reducing infection rates have become a public health priority [56].

To conclude, CCA is a complex and highly aggressive malignancy with distinct subtypes depending on the anatomical origin of the tumor within the bile ducts. The risk factors for CCA include primary sclerosing cholangitis, liver fluke infection, cirrhosis, and chronic bile duct injury, with pathogenesis rooted in chronic inflammation, bile stasis, and genetic mutations (Figure 1). Despite the differences between iCCA and eCCA in terms of clinical presentation and molecular characteristics, both forms carry a poor prognosis, highlighting the need for better diagnostic tools and therapeutic interventions. Understanding the underlying mechanisms of CCA pathogenesis and the risk factors that predispose individuals to this malignancy is critical for developing more effective preventive and treatment strategies.

##### Genetic and Genomic Alterations in Cholangiocarcinoma

CCA, particularly iCCA, displays distinct genetic features compared to HCC, with a high prevalence of *IDH1/IDH2*, *FGFR2*, *BAP1*, and *ARID1A* mutations. These alterations define unique molecular subtypes of CCA with implications for targeted therapy [57,58,59].

Mutations in *IDH1/2*, which occur in 10–20% of iCCA cases, result in aberrant production of 2-hydroxyglutarate, a metabolite that alters the epigenome and impairs cellular differentiation [59]. *FGFR2* fusions are another hallmark of iCCA, particularly in non-cirrhotic livers, and have led to the development of specific *FGFR* inhibitors now approved for clinical use [60].

Loss-of-function mutations in *BAP1*, a tumor suppressor involved in chromatin remodeling, are associated with poor prognosis [61]. *ARID1A* and *PBRM1*, members of the SWI/SNF chromatin remodeling complex, are also frequently altered in CCA, indicating a disruption in epigenetic control mechanisms [62].

In contrast, extrahepatic CCA is more often associated with *KRAS*, *SMAD4*, and *TP53* mutations, reflecting a molecular profile more similar to pancreatic and colorectal cancers. These differences highlight the need to stratify patients by CCA subtype for precision oncology [63,64].

Inflammatory conditions such as primary sclerosing cholangitis and liver fluke infections promote chronic bile duct inflammation, leading to an accumulation of genetic mutations. Moreover, evidence suggests that dysbiosis and microbial byproducts may indirectly contribute to mutagenesis via oxidative stress and DNA methylation changes. Frequently mutated genes in HCC and CCA, along with their functional roles and associated references, are summarized in Table 1.

### 3.2. The Role of Microbiota in Hepatic Cancer Development

#### 3.2.1. Microbiota and Hepatocellular Carcinoma

The interplay between gut microbiota and HCC is increasingly recognized as a significant factor in the pathogenesis of this aggressive form of liver cancer [65]. The gut and liver are closely connected through the gut–liver axis, a bidirectional communication pathway that influences various physiological processes [65]. In the context of liver diseases, particularly chronic liver disease and cirrhosis, alterations in the gut microbiota, referred to as dysbiosis, can create a pro-carcinogenic environment that promotes HCC development [65].

##### Dysbiosis and Cirrhosis in HCC Development

Chronic liver disease, including conditions such as hepatitis B and C infections, alcohol-related liver disease, and NAFLD, often progresses to cirrhosis. This state of chronic liver injury significantly disrupts the homeostasis of gut microbiota [66]. Dysbiosis is characterized by an imbalance in microbial populations, where pathogenic species proliferate at the expense of beneficial microbes [66]. In patients with cirrhosis, studies have demonstrated significant shifts in gut microbial composition. For instance, there is often an increase in Gram-negative bacteria (e.g., Enterobacteriaceae) and a decrease in beneficial genera such as Lactobacillus and Bifidobacterium [66]. These changes are not merely coincidental; they are driven by factors such as impaired liver function, portal hypertension, and increased intestinal permeability. The compromised gut barrier allows for the translocation of bacteria and their products into systemic circulation, leading to a phenomenon often described as “leaky gut” [66,67]. This translocation is particularly concerning, as it facilitates the entry of lipopolysaccharides (LPS)—components of the outer membrane of Gram-negative bacteria—into the bloodstream, triggering systemic inflammation [67]. The liver, being the primary organ responsible for clearing these microbial products, becomes chronically exposed to LPS, which activates the innate immune response [66,67]. This inflammatory microenvironment can lead to hepatic injury, fibrosis, and ultimately, a conducive environment for carcinogenesis [66,67]. Research has indicated that alterations in the gut microbiota composition are not only prevalent in cirrhosis but also correlate with the progression to HCC. For instance, studies have found that specific microbial signatures are associated with HCC in patients with underlying cirrhosis, highlighting the potential of microbiota alterations as biomarkers for early detection and risk stratification in liver cancer [66,67].

##### Metabolites and Cancer Promotion

The gut microbiota produces a diverse array of metabolites that can significantly impact liver health, influencing both protective and harmful processes. Among these, secondary bile acids and SCFAs have garnered attention for their roles in liver carcinogenesis [68,69]. Secondary bile acids are formed from primary bile acids under the action of gut microbiota [68,70]. In the context of dysbiosis, the profile of bile acids can shift dramatically, leading to the accumulation of potentially harmful secondary bile acids such as deoxycholic acid (DCA). DCA has been shown to promote hepatocyte apoptosis, induce DNA damage, and facilitate the development of a pro-inflammatory microenvironment, all of which are key factors in the progression from cirrhosis to HCC [68,69,70]. Additionally, altered bile acid profiles can activate specific nuclear receptors in the liver, such as farnesoid X receptor (FXR), which are involved in liver inflammation and metabolism, thereby contributing to a carcinogenic environment [68,69,70]. SCFAs, primarily produced through the fermentation of dietary fibers by gut bacteria, are generally associated with protective effects on liver health due to their anti-inflammatory properties and ability to strengthen the gut barrier [71,72]. However, in states of dysbiosis, the production of SCFAs may be diminished, exacerbating gut permeability and fostering a pro-inflammatory state [71,72]. Moreover, SCFAs can modulate immune responses in the liver, and their deficiency can lead to an imbalance in immune cell populations, further promoting liver inflammation and cancer development [71,72].

##### Chronic Inflammation and Immune Modulation

The gut microbiota is crucial in shaping liver inflammation and modulating immune responses, particularly in the context of chronic liver diseases and HCC [65,66,67,68]. The intricate relationship between the gut and liver means that dysbiosis can profoundly impact immune regulation in the liver [72]. Chronic exposure to microbial products, especially LPS, activates pattern recognition receptors (PRRs) on immune cells in the liver, primarily Kupffer cells, leading to the production of a range of pro-inflammatory cytokines, such as IL-6, TNF-α, and IL-1β [73,74,75]. IL-6 is particularly noteworthy in the context of HCC, as it promotes hepatocyte proliferation and survival while inhibiting apoptosis [73,76]. Elevated levels of IL-6 have been associated with worse prognoses in patients with HCC, emphasizing its role as a key player in liver cancer progression [73,76]. Similarly, TNF-α contributes to inflammation and liver damage, creating an environment that supports the development of HCC [74]. Moreover, the chronic inflammatory state fostered by dysbiosis leads to immune system dysregulation. The balance between pro-inflammatory and anti-inflammatory immune responses is critical for maintaining liver health. In chronic liver disease, there is often an increase in regulatory T cells (Tregs) and myeloid-derived suppressor cells (MDSCs), which suppress anti-tumor immunity and promote tumor progression. This immune evasion allows cancer cells to thrive despite the presence of an inflammatory environment [77,78]. Furthermore, chronic inflammation can lead to the activation of fibrogenic pathways, resulting in fibrosis and cirrhosis, which are established risk factors for HCC. In summary, the gut microbiota significantly influences liver inflammation and immune responses, with dysbiosis contributing to a cycle of chronic inflammation, immune dysregulation, and ultimately, the promotion of hepatocellular carcinoma. Understanding these mechanisms offers valuable insights into potential therapeutic strategies targeting the gut–liver axis to prevent and treat HCC.

#### 3.2.2. Microbiota and Cholangiocarcinoma

The role of gut microbiota in the development of CCA, a malignancy arising from the bile ducts, is a growing area of research. CCA is characterized by a complex interplay between biliary dysbiosis, bile acid metabolism, and gut barrier dysfunction [8,9,10]. Understanding how these factors contribute to cholangiocarcinogenesis is crucial for developing preventive and therapeutic strategies against this aggressive cancer.

##### Biliary Dysbiosis

The biliary tree, comprising the intrahepatic and extrahepatic bile ducts, harbors a unique microbiota composition distinct from that of the gut [70]. While traditionally considered sterile, recent studies have identified a diverse microbial community within the bile ducts, which can be altered by various factors, including inflammation, infection, and metabolic changes [44,79]. Dysbiosis in this niche disrupts the ecological balance, increasing susceptibility to biliary diseases such as cholangiocarcinoma (CCA) [44,79]. A well-established infectious cause of biliary dysbiosis is liver fluke infection—particularly *Opisthorchis viverrini* and *Clonorchis sinensis*—which contributes to chronic inflammation and has been strongly associated with CCA development in endemic areas [80].

However, non-infectious contributors to biliary dysbiosis are increasingly recognized as important in the pathogenesis of CCA, especially in non-endemic regions. Factors such as altered bile acid metabolism, host immune dysregulation, antibiotic use, diet, and environmental toxins can independently shape the biliary microbiome [18,81,82]. For example, chronic cholestasis, primary sclerosing cholangitis (PSC), and non-alcoholic fatty liver disease (NAFLD) can modify bile flow and composition, influencing microbial colonization and selection within the bile ducts. These non-infectious conditions promote a pro-inflammatory and fibrotic microenvironment that favors microbial overgrowth and pathogenic colonization. Dysbiosis in these contexts has been associated with increased levels of pro-inflammatory bacteria (e.g., *Enterococcus*, *Escherichia coli*, and *Fusobacterium*) and reduced microbial diversity, both of which are linked to carcinogenesis.

Additionally, dysregulated bile acid signaling—such as through altered FXR or TGR5 pathways—can impair epithelial defense mechanisms and barrier function, further facilitating microbial shifts and biliary injury [81]. Cancer therapies and frequent antibiotic exposure in cancer patients may also lead to unintended disruption of the biliary and gut microbiomes, thereby perpetuating inflammation and increasing vulnerability to malignancy.

As in the case of hepatocellular carcinoma (HCC), dysbiosis affects not only local microbial populations but also gut–liver axis integrity. A compromised gut barrier can allow translocation of microbial products into the portal circulation, exacerbating hepatic and biliary inflammation. This so-called “leaky gut” phenomenon contributes to a pro-oncogenic microenvironment via activation of Toll-like receptors and release of cytokines and reactive oxygen species [67].

Altogether, both infectious and non-infectious drivers of biliary dysbiosis contribute to the multifactorial landscape of cholangiocarcinogenesis, highlighting the importance of microbial homeostasis in maintaining bile duct health and preventing malignant transformation.

#### 3.2.3. Host Genetic Variation and Microbiota Interactions in Hepatic Carcinogenesis

Recent studies have highlighted the bidirectional relationship between host genetics and microbial ecology, emphasizing how genetic variation can influence microbiota composition, and conversely, how microbial metabolites and products can impact genomic stability and cancer development in the liver. Genome-wide association studies (GWAS) and host–microbiome profiling have revealed that genetic polymorphisms in immune and metabolic genes, such as *TLR4*, *NOD2*, *MYD88*, and *FUT2*, can significantly alter the abundance and function of key microbial taxa [83,84,85,86]. These genes regulate processes like pathogen recognition, bile acid metabolism, and intestinal barrier integrity, all of which play critical roles in maintaining a balanced gut microbiome. In individuals with susceptible genetic variants, these pathways may be impaired, leading to microbial dysbiosis [83,84,85,86]. Dysbiosis, in turn, fosters increased intestinal permeability and facilitates translocation of microbial products—such as lipopolysaccharides (LPS) and peptidoglycans—into the portal circulation, triggering chronic hepatic inflammation and a tumor-promoting microenvironment [52,53].

At the same time, microbial metabolites such as deoxycholic acid (DCA), short-chain fatty acids (SCFAs), trimethylamine-N-oxide (TMAO), and reactive oxygen species (ROS) can promote oxidative DNA damage, impair DNA repair mechanisms, and alter epigenetic landscapes in hepatocytes [87,88,89]. These effects may increase the accumulation of mutations in genes commonly altered in hepatic cancers, including *TP53*, *TERT*, *ARID1A*, and *CTNNB1* [41,42,46]. Chronic exposure to genotoxic microbial products can thus potentiate genomic instability and accelerate tumorigenesis, particularly in the context of a genetically primed immune or metabolic state.

Moreover, host–microbiota interactions significantly shape hepatic immune responses. Polymorphisms in genes encoding cytokines (*IL6*, *TNF*, and *IL10*) and HLA molecules influence the liver’s immunological tolerance and tumor surveillance capacity [90,91]. These variations can modulate how microbial signals are processed by Kupffer cells, dendritic cells, and other hepatic immune subsets, thereby affecting inflammation thresholds and the effectiveness of anti-tumor immunity. This has direct relevance for immunotherapy, as emerging data suggest that both host genotypes and microbial composition influence patient responses to immune checkpoint inhibitors in liver cancer.

Integration of host genomics with metagenomic and transcriptomic data is rapidly advancing our understanding of this multilayered network. These systems biology approaches are uncovering how specific genetic variants interact with microbial features to drive liver disease progression, cancer risk, and therapeutic outcomes. Ultimately, characterizing the host–microbiota–genome axis offers a promising avenue for precision oncology, enabling personalized risk assessment, early detection strategies, and tailored interventions that consider both the genetic background and microbial environment of liver cancer patients.

### 3.3. Microbiota as a Diagnostic and Prognostic Tool in Hepatic Cancers

The increasing understanding of the gut–liver axis has led to the recognition of microbiota as a potential diagnostic and prognostic tool in hepatic cancers, including HCC and CCA. Microbial composition and diversity are being investigated for their roles in early cancer detection and the prediction of treatment responses, providing a promising avenue for improving patient management.

#### 3.3.1. Microbiota Profiling for Diagnosis

Recent studies have explored the gut microbiota composition as a biomarker for the early detection of HCC and CCA, emphasizing its potential as a non-invasive diagnostic tool. Research has demonstrated significant differences in the gut microbial profiles of patients with hepatic cancers compared to healthy individuals [92]. For instance, specific bacterial taxa, such as Alistipes or Bacteroides, have been found to be overrepresented in patients with HCC, while other genera may be underrepresented [93,94]. These microbial alterations reflect underlying pathophysiological changes associated with cancer development and progression, making them valuable for diagnostic purposes.

Emerging tools such as metagenomics and next-generation sequencing (NGS) have revolutionized the ability to identify microbial signatures in hepatic cancer patients [94]. Metagenomics allows for the comprehensive analysis of microbial DNA from fecal samples, enabling the identification of both known and novel microbial species [94]. NGS facilitates high-throughput sequencing, providing detailed insights into microbial diversity and composition with unprecedented resolution.

However, despite these promising developments, the utility of microbiota profiling as a diagnostic tool must be interpreted cautiously. Microbiota composition can be influenced by numerous confounding factors, including geography, diet, age, medication use (e.g., antibiotics), and the choice of sequencing platform and bioinformatic pipeline [95,96,97]. These variables can introduce bias or variability into microbial datasets, potentially limiting the generalizability of findings across diverse populations.

#### 3.3.2. Microbiota and Cancer Prognosis

In addition to diagnostic applications, the composition of gut microbiota is emerging as a crucial factor in predicting the prognosis of liver disease patients [18,98]. Recent investigations—primarily based on human cohort studies—have established associative correlations between specific microbiota profiles and patient outcomes, suggesting that microbial signatures may serve as prognostic biomarkers in HCC and CCA [98]. For example, certain microbial taxa associated with anti-inflammatory properties have been linked to improved survival rates, while dysbiotic profiles characterized by pro-inflammatory bacterial species have been associated with poorer outcomes [98]. However, these correlations do not establish direct causality, and mechanistic evidence from preclinical models remains limited.

Moreover, the gut microbiota may influence the response to cancer therapies, including immunotherapy and chemotherapy [22,99,100]. Several studies using animal models and in vitro systems have begun to elucidate potential mechanisms, including microbial modulation of drug metabolism and immune pathways [90,91]. Nonetheless, most clinical data to date are observational, and the directionality of these relationships in humans remains unclear. Conversely, dysbiosis may hinder treatment efficacy, leading to treatment resistance and suboptimal patient outcomes [101,102]. For instance, research has indicated that certain microbial populations can metabolize chemotherapeutic agents, potentially altering their effectiveness [101,102].

Therefore, while the prognostic potential of microbiota is promising, caution is warranted in interpreting these findings. Distinguishing between correlation and causation is crucial, particularly given that much of the existing evidence is derived from observational studies in human cohorts rather than from controlled, mechanistic investigations. Ongoing preclinical research and longitudinal, intervention-based clinical studies will be essential to validate causal roles and therapeutic implications.

As with diagnostic studies, prognostic microbiota analyses must also account for confounders such as host genetics, environmental exposures, and sample processing techniques, which may affect microbial community structure and function [95,96,97]. Rigorous standardization and validation in larger, multi-center studies will be essential for translating these findings into clinical practice.

### 3.4. Therapeutic Potential of Targeting Microbiota in Hepatic Cancers

The therapeutic potential of targeting gut microbiota in the context of hepatic cancers is an emerging field that offers innovative approaches for managing diseases such as HCC and CCA [17,18,23]. As our understanding of the gut–liver axis deepens, various strategies aimed at modulating the gut microbiome are being explored to enhance anti-carcinogenic effects and improve patient outcomes. This section delves into several promising interventions, including probiotics, prebiotics, synbiotics, fecal microbiota transplantation, the impact of microbiota on immune checkpoint inhibitors, and dietary modulation of the microbiome.

#### 3.4.1. Probiotics, Prebiotics, and Synbiotics

Probiotics, which are live microorganisms that confer health benefits; prebiotics, which are non-digestible food components that promote the growth of beneficial bacteria; and synbiotics, which combine both probiotics and prebiotics, hold significant potential for modulating gut microbiota in favor of anti-carcinogenic effects [103]. Research has shown that these interventions can positively influence microbial composition [103,104,105], enhance gut barrier function, and reduce inflammation, all of which are crucial for preventing the progression of liver diseases to cancer. Experimental and clinical studies have explored the efficacy of these interventions in patients with HCC and chronic liver diseases [106,107]. For instance, a clinical trial investigated the administration of specific probiotic strains in patients with cirrhosis, revealing improvements in liver function markers and a reduction in complications related to liver disease [108]. Similarly, prebiotic supplementation helps to prevent dysregulated microbial fermentation of soluble fiber, which is generally correlated with liver disease [71,109]. In the context of liver cancer, synbiotics have shown promise in preclinical studies by exhibiting synergistic effects that improve microbial diversity and modulate immune responses [110]. Synbiotic treatment has a role in mitigating chemoradiotherapy-associated toxicity in colorectal cancer patients, highlighting its potential to improve treatment outcomes and reduce adverse side effects [110], warranting further exploration in larger clinical trials.

#### 3.4.2. Fecal Microbiota Transplantation

Fecal microbiota transplantation (FMT) is gaining attention as a novel therapeutic approach for treating liver diseases and has potential applications in hepatic cancers [111,112,113,114]. FMT involves transferring fecal material from a healthy donor to the gastrointestinal tract of a patient, aiming to restore a healthy microbial composition [111]. FMT has emerged as a promising therapeutic option for cirrhosis and liver cancer treatment by restoring gut microbial balance and improving liver health [112,113]. In cirrhosis, gut dysbiosis contributes to systemic inflammation, immune dysfunction, and the progression of liver disease, often leading to complications like hepatic encephalopathy [115,116]. FMT can help replenish beneficial gut bacteria, reducing systemic inflammation, enhancing gut barrier function, and potentially slowing disease progression [115,116]. In liver disease, particularly NAFLD and HCC, FMT may enhance treatment efficacy by modulating the gut–liver axis, improving immune responses, and reducing pro-inflammatory cytokines that contribute to tumor growth [82,117]. Preliminary studies suggest that FMT may also improve patient outcomes in combination with immunotherapy by enhancing the immune system’s response to cancer [112,113], though more research is needed to confirm its long-term benefits in this context.

#### 3.4.3. Microbiota and Immune Checkpoint Inhibitors

The interaction between gut microbiota and the efficacy of immune checkpoint inhibitors (ICIs), such as anti-PD-1 and anti-PD-L1 therapies, represents a critical area of research in the management of hepatic cancers [92,118]. ICIs have revolutionized cancer treatment, but their effectiveness varies significantly among patients. Recent studies suggest that the composition of gut microbiota may play a pivotal role in determining the response to these therapies [119,120]. While most of the current mechanistic evidence comes from studies in melanoma and other solid tumors, emerging data in hepatocellular carcinoma (HCC) support a potential link between the microbiome and ICI response. For example, altered gut microbial diversity and enrichment of specific taxa—such as *Akkermansia muciniphila* and Ruminococcaceae—have been associated with better ICI responses in HCC patients receiving anti-PD-1 therapy [120,121,122,123,124]. Moreover, fecal microbiota transplantation (FMT) from ICI-responsive donors has been shown to enhance antitumor immunity in murine HCC models, suggesting a causal relationship between gut microbial composition and immunotherapy efficacy [120,121,122,123,124].

Although comparable data in cholangiocarcinoma (CCA) are currently limited, ongoing trials are investigating how the gut–liver axis and immune modulation affect therapeutic outcomes in this setting [122,123,124,125]. The microbiome’s role in influencing immunotherapy outcomes remains an active area of research in hepatic cancers.

In other malignancies, such as metastatic melanoma, patients with a higher abundance of certain bacteria—such as *Bifidobacterium longum*, *Collinsella aerofaciens*, and *Enterococcus faecium*—have shown better clinical responses [119]. Additionally, transferring fecal material from responding patients into germ-free mice improved tumor control and enhanced anti-PD-L1 therapy efficacy, indicating a possible mechanistic link between the microbiome and immune response [119].

### 3.5. Future Directions and Research Gaps

As the understanding of the gut–liver axis and the role of microbiota in hepatic cancers continues to evolve, several future directions and research gaps become evident. Addressing these areas is critical for translating basic research findings into clinical applications that can improve the diagnosis, treatment, and prevention of liver cancers such as HCC and CCA.

#### 3.5.1. Mechanistic Studies

One of the most significant gaps in current research is the need for mechanistic studies that explore the causal relationship between microbiota alterations and hepatic cancer development. While observational studies have established correlations between dysbiosis and liver cancer, the underlying mechanisms remain largely unexplored [23,65,66,67,68]. Mechanistic studies can clarify how specific microbial species or community structures influence liver pathology, inflammation, and carcinogenesis. To advance this area of research, various approaches can be employed, including the use of animal models and organoids. Animal models, particularly genetically modified mice and germ-free systems, provide valuable insights into how alterations in gut microbiota can directly affect liver function and cancer progression [121,122,123]. For instance, introducing specific bacterial strains into germ-free mice and observing changes in liver tumor development can help elucidate the roles of particular microbes in hepatic carcinogenesis [121,122,123]. Organoids, which are three-dimensional cultures derived from human tissues, represent another promising tool [124,125]. By cultivating liver organoids alongside different microbiota profiles, researchers can study the interactions between gut-derived signals and liver cells in a controlled environment, providing insights into the mechanisms by which microbiota influence liver health and disease [124,125].

#### 3.5.2. Longitudinal Studies

Longitudinal studies are crucial for assessing how changes in microbiota over time relate to cancer risk, progression, and response to therapies. These studies involve monitoring the same individuals over extended periods, allowing researchers to track microbiome dynamics in relation to various clinical events, such as the onset of liver disease or the initiation of cancer therapies [126,127]. By establishing baseline microbiome profiles and subsequently tracking changes, researchers can identify specific microbial shifts that may precede or accompany disease progression. For instance, early detection of dysbiosis could serve as a predictive marker for the development of HCC or CCA, enabling earlier intervention and potentially improving outcomes. Longitudinal studies also provide insights into the impact of therapeutic interventions on microbiota composition and function [128,129,130]. By evaluating how different treatments, including chemotherapy, immunotherapy, or dietary changes, influence gut microbiota over time, researchers can better understand the interplay between treatment efficacy and microbial health. This knowledge could inform the timing and selection of therapeutic strategies to maximize benefits while minimizing adverse effects.

In conclusion, addressing these future directions and research gaps is vital for harnessing the therapeutic potential of microbiota in the context of hepatic cancers. Mechanistic studies will provide insights into the causal relationships between microbiota and liver cancer, while personalized microbiome-based therapies and longitudinal studies will pave the way for innovative treatment approaches that enhance patient outcomes. By integrating microbiome research into clinical practice, we can significantly improve our understanding and management of hepatic cancers, ultimately leading to better prognoses and quality of life for patients.

## 4. Discussion

This review highlights the intricate relationship between gut microbiota and hepatic cancers, specifically focusing on HCC and CCA (Figure 2). It underscores how alterations in gut microbiota composition, termed dysbiosis, can significantly impact liver health and contribute to the pathogenesis of these malignancies. The evidence suggests that dysbiosis not only exacerbates chronic liver conditions such as cirrhosis but also plays a crucial role in promoting inflammation and carcinogenesis through various mechanisms, including immune modulation and the production of microbial metabolites.

Moreover, the potential of microbiota as a diagnostic and therapeutic avenue in hepatobiliary cancers is promising. Advances in microbiota profiling using cutting-edge techniques like metagenomics and NGS have paved the way for identifying microbial signatures associated with HCC and CCA. These microbiota profiles could serve as biomarkers for early detection and prognosis, enabling personalized treatment strategies that take into account individual variations in gut microbial communities. Interventions aimed at modulating the microbiota, such as probiotics, prebiotics, synbiotics, and fecal microbiota transplantation, hold potential for enhancing therapeutic efficacy and improving patient outcomes in hepatic cancers.

However, while the current findings are encouraging, further research is essential to fully elucidate the complex interplay between microbiota and hepatic cancer. Mechanistic studies are needed to establish causal relationships and understand the underlying pathways through which microbiota influence liver carcinogenesis. Additionally, longitudinal studies can provide insights into the dynamic nature of microbiota changes over time and their implications for cancer risk and treatment response.

In conclusion, as the field of microbiome research continues to expand, it is imperative to focus on integrating microbiota considerations into clinical practice for hepatic cancers. By leveraging the potential of microbiota as both diagnostic tools and therapeutic targets, we may improve early detection, enhance treatment strategies, and ultimately achieve better clinical outcomes for patients with liver cancer.

## Figures and Tables

**Figure 1 genes-16-00920-f001:**
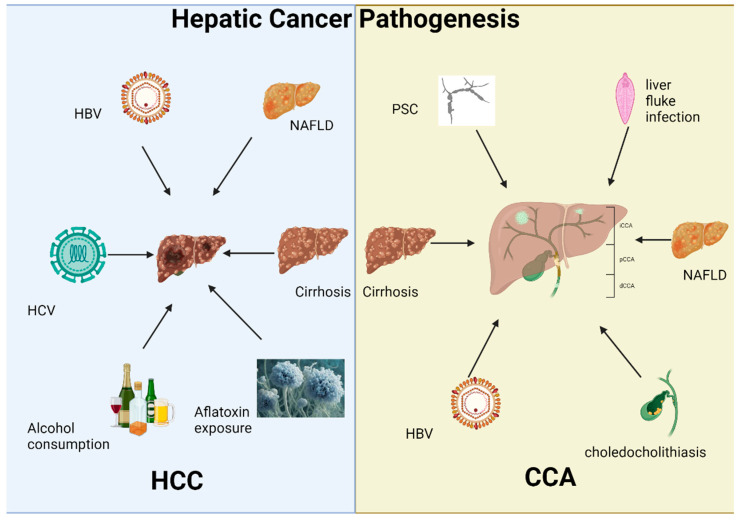
Overview of the pathogenesis of HCC and CCA. The left panel delineates the principal risk factors for hepatocellular carcinoma (HCC), including chronic viral hepatitis (HBV and HCV), excessive alcohol consumption, non-alcoholic fatty liver disease (NAFLD), cirrhosis, and aflatoxin exposure. Conversely, the right panel outlines the predominant risk factors for cholangiocarcinoma (CCA), which comprise primary sclerosing cholangitis (PSC), chronic liver fluke infections, cirrhosis, NAFLD, and choledocholithiasis.

**Figure 2 genes-16-00920-f002:**
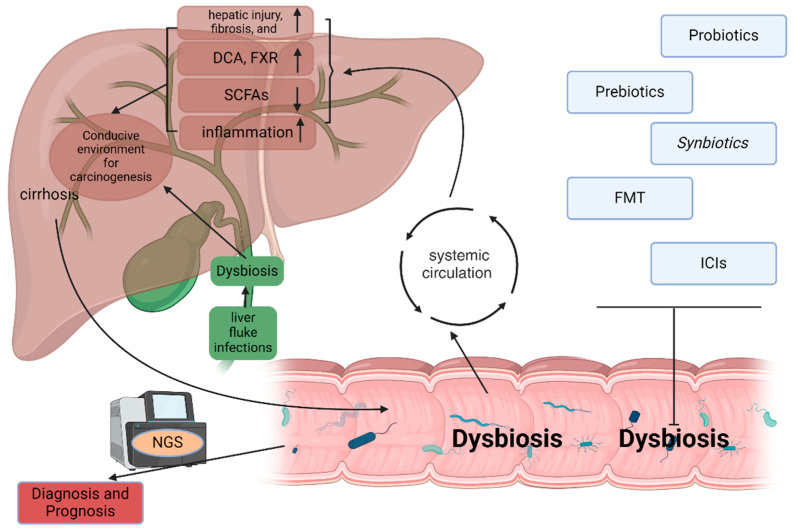
Outline of the influence of microbiota in HCC and CCA. Chronic liver disease frequently progresses to cirrhosis, which disrupts gut microbiota homeostasis. This dysbiosis facilitates bacterial translocation into the systemic circulation. Translocated microbial components subsequently induce hepatic inflammation, exacerbate hepatic injury, and accelerate fibrogenesis. Concurrently, altered microbial metabolism elevates carcinogenic compounds such as deoxycholic acid (DCA) while suppressing protective short-chain fatty acids (SCFAs) and impairing farnesoid X receptor (FXR) signaling. These collective mechanisms establish a microenvironment conducive to hepatocarcinogenesis. Furthermore, biliary-specific dysbiosis—notably induced by chronic liver fluke infections—heightens susceptibility to malignancies such as cholangiocarcinoma. The characterized gut dysbiosis can be used for diagnosis and prognosis of HCC and CAA via NSG. Additionally, the therapeutic potential of targeting gut microbiota could be realized via robiotics, prebiotics, synbiotics, fecal microbiota transplantation (FMT), and immune checkpoint inhibitors (ICIs).

**Table 1 genes-16-00920-t001:** Frequently mutated genes in HCC and CCA.

Gene	Function/Role	Cancer Type	Mutation Frequency/Relevance	References
*TP53*	Tumor suppressor, DNA repair, apoptosis	HCC, CCA	~30–50% in HCC; also common in eCCA	[45,63]
*CTNNB1*	Wnt/β-catenin pathway activation	HCC	Activating mutations that drive proliferation	[40,46]
*TERT*	Telomerase reverse transcriptase	HCC, CCA	Promoter mutations in early stages of HCC	[33,42,47]
*IDH1/2*	Metabolism, epigenetic modulation	iCCA	~10–20% in iCCA, rare in HCC	[57,59]
*FGFR2*	Tyrosine kinase receptor, cell growth	iCCA	Fusions/rearrangements targetable by inhibitors	[60]
*ARID1A*	Chromatin remodeling	HCC, CCA	Frequently mutated; epigenetic dysregulation	[48,62]
*BAP1*	Tumor suppressor, chromatin regulation	CCA	Loss-of-function linked to poor prognosis	[61]
*KRAS*	RAS signaling pathway	eCCA	Commonly mutated in eCCA, less frequent in iCCA or HCC	[63]
*SMAD4*	TGF-β signaling, tumor suppression	eCCA	Frequently altered in eCCA	[63]
*AXIN1*	Wnt pathway inhibitor	HCC	Inactivation promotes β-catenin signaling	[48]
*KEAP1/NFE2L2*	Oxidative stress response	HCC	Mutations impair ROS detox pathways	[48]
*PBRM1*	SWI/SNF complex, chromatin regulation	CCA	Frequently mutated; epigenetic dysregulation	[62]
*MLL3/KMT2C*	Histone methyltransferase	HCC	Mutated in HCC; linked to tumor suppressor CDKN2A	[50]
*CDKN2A*	Cell cycle regulation	HCC, CCA	Methylated or deleted in both cancers	[49,50]
*RASSF1A*	Tumor suppressor	HCC	Silenced via promoter methylation	[42]

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
