# Peer review of "Genomic Alterations and Microbiota Crosstalk in Hepatic Cancers: The Gut–Liver Axis in Tumorigenesis and Therapy"

_genes, 2025, doi:10.3390/genes16080920_

Round 1
Reviewer 1 Report
Comments and Suggestions for Authors
Dear Authors,
The article submitted for review comprehensively presents current knowledge on the genetic and genomic drivers of hepatic cancers, with the role of gut microbiota emphasizing their bidirectional interactions. The article is written in an excellent, reader-friendly manner. The authors fully achieved the intended goal of the work. Below are my minor comments:
1. Abbreviations used in figures should be described to facilitate reader understanding of the content.
2. The rather innovative short discussion, without references to other works (references), serves more of a summary function. The discussion concludes with brief conclusions that are worth highlighting.
Other than that, as I mentioned, I have no major comments.
Author Response
The article submitted for review comprehensively presents current knowledge on the genetic and genomic drivers of hepatic cancers, with the role of gut microbiota emphasizing their bidirectional interactions. The article is written in an excellent, reader-friendly manner. The authors fully achieved the intended goal of the work. Below are my minor comments:
- Abbreviations used in figures should be described to facilitate reader understanding of the content.
Thank you very much for your comment. We have corrected all the figure captions for clarity. - The rather innovative short discussion, without references to other works (references), serves more of a summary function. The discussion concludes with brief conclusions that are worth highlighting.
We thank the reviewer for such positive comments. Since this is a review, we wanted to keep the discussion brief and without references so as not to overload the reader, since we believe that a discussion section of this nature is more interesting in a review than a more “classic” discussion, such as the one that is usually written in a research article. We are pleased to know that the reviewer found it interesting and agrees with our thinking.

Reviewer 2 Report
Comments and Suggestions for Authors
This manuscript addresses an important topic by summarizing current knowledge regarding the incidence, molecular contributors, and potential therapeutic strategies for hepatocellular carcinoma (HCC) and cholangiocarcinoma (CCA). The subject is timely and of clinical significance. However, the manuscript suffers from substantial issues related to redundancy, organizational clarity, and structure, which collectively diminish its readability and scientific impact. I strongly recommend the following revisions to improve the manuscript’s clarity and credibility:
- The manuscript title contains a typographical error with two consecutive uses of the word “in.” Please revise for clarity and correctness.
- The introduction is verbose and repetitive; for example, global incidence statistics (lines 36 and 68) and statements about the disease burden of HCC and CCA are repeated multiple times.
- I recommend a concise rewrite that begins with the global burden of hepatic cancers, followed by a clear rationale for focusing on HCC and CCA, a brief explanation of their cellular origins, and the relevance of gut microbiota in this context.
- Avoid sub-sectioning within the introduction unless absolutely necessary. A cohesive, well-structured narrative would improve clarity and flow.
- There is a discontinuity in the numbering of introduction subsections, with section 1.1 followed directly by section 1.3. Please revise to ensure consistent section numbering and logical flow.
- The objective section is currently too descriptive. Consider listing only the key objectives as bullet points and omitting any extended explanation, which can be expanded upon later in the manuscript body.
- The results section opens with information already covered in the introduction, including the incidence of HCC and common gene mutations. Such redundancy should be avoided. Streamline the content to ensure each point is addressed in full only once, and where most appropriate.
- Figure 1 is not referenced anywhere in the main text. Please ensure all figures are appropriately cited and integrated into the narrative
- I recommend adding a comprehensive table summarizing frequently mutated genes in HCC and CCA, along with corresponding references. This will enhance the utility of the review for readers.
Author Response
This manuscript addresses an important topic by summarizing current knowledge regarding the incidence, molecular contributors, and potential therapeutic strategies for hepatocellular carcinoma (HCC) and cholangiocarcinoma (CCA). The subject is timely and of clinical significance. However, the manuscript suffers from substantial issues related to redundancy, organizational clarity, and structure, which collectively diminish its readability and scientific impact. I strongly recommend the following revisions to improve the manuscript’s clarity and credibility:
- The manuscript title contains a typographical error with two consecutive uses of the word “in.” Please revise for clarity and correctness.
Thank you very much for your comment. We apologize for this error. It has been corrected.
- The introduction is verbose and repetitive; for example, global incidence statistics (lines 36 and 68) and statements about the disease burden of HCC and CCA are repeated multiple times.
- I recommend a concise rewrite that begins with the global burden of hepatic cancers, followed by a clear rationale for focusing on HCC and CCA, a brief explanation of their cellular origins, and the relevance of gut microbiota in this context.
- Avoid sub-sectioning within the introduction unless absolutely necessary. A cohesive, well-structured narrative would improve clarity and flow.
Thank you very much for your comments. In relation to these 3 comments regarding the introduction, the entire introduction has been rewritten, eliminating redundant statistics and repeated statements about disease burden and presenting a cohesive, linear narrative without subsections, achieving greater clarity, conciseness, and structure.
- There is a discontinuity in the numbering of introduction subsections, with section 1.1 followed directly by section 1.3. Please revise to ensure consistent section numbering and logical flow.
In accordance with the previous comment, the subsections of the introduction have been eliminated, so this problem has been corrected.
- The objective section is currently too descriptive. Consider listing only the key objectives as bullet points and omitting any extended explanation, which can be expanded upon later in the manuscript body.
Thank you very much for your comments. The objective section has been removed and included in the body of the manuscript.
- The results section opens with information already covered in the introduction, including the incidence of HCC and common gene mutations. Such redundancy should be avoided. Streamline the content to ensure each point is addressed in full only once, and where most appropriate.
Thank you very much for your comment. We have deleted the following paragraph to eliminate redundancy with respect to the introduction, keeping the following parts, which provide new information: In areas such as East Asia and sub-Saharan Africa, chronic HBV infections represent the major cause of HCC [2,7]. Conversely, in Western nations, HCV, alcohol consumption, and NAFLD emerge as the primary risk factors for this type of cancer [2,3].
- Figure 1 is not referenced anywhere in the main text. Please ensure all figures are appropriately cited and integrated into the narrative
We apologize for this error. We have included the citations corresponding to Figure 1 in the text.
- I recommend adding a comprehensive table summarizing frequently mutated genes in HCC and CCA, along with corresponding references. This will enhance the utility of the review for readers.
Table 1 has been created and introduced in the text, following the reviewer's indications.

Reviewer 3 Report
Comments and Suggestions for Authors
This review article by Fu et al. explores the intricate relationship between the gut microbiota and hepatic cancers, focusing on hepatocellular carcinoma (HCC) and cholangiocarcinoma (CCA). It synthesizes current literature on genetic and epigenetic alterations, microbiota-driven inflammatory and metabolic changes, and host–microbe interactions that influence hepatic tumorigenesis. The review also evaluates the diagnostic and therapeutic potential of microbiota modulation and outlines gaps in the field. The manuscript is comprehensive, well-organized, and addresses a timely and increasingly relevant research area, with strong potential appeal to both basic scientists and clinicians.
Specific Comments
- Line 348: The figure legend for Figure 1 is missing any explanatory detail beyond the title. A brief description of key pathways illustrated would enhance clarity.
- Lines 427–458: The coverage of microbiota’s role in CCA is less comprehensive than for HCC. While liver fluke–associated CCA is well-described, further discussion of non-infectious dysbiosis in CCA would be helpful.
- Lines 498–533: The enthusiasm about microbiota as a diagnostic/prognostic tool is appropriate and well-timed, but a brief mention of confounding factors, such as population, diet, sequencing platform would add necessary balance.
- Lines 503–533: The prognostic role of the microbiota is well discussed, but the section would benefit from clearer distinction between correlation and causation. This is especially important because these findings are derived from human cohort studies rather than preclinical/mechanistic studies. Consider flagging which data come from human cohorts vs. animal models.
- Line 563: The statement that FMT has potential applications in “other age-related diseases” is out of scope unless the authors briefly elaborate or remove this phrase.
- Line 580–593: The section on immune checkpoint inhibitors focuses largely on melanoma studies. Consider briefly discussing what (if anything) is known in HCC/CCA models or clinical cohorts.
- Lines 594–641: Strong section overall. Consider citing specific examples where liver organoid-microbiota co-cultures have been used, if available.
- Line 671: Figure 2 legend needs to be expanded to describe everything depicted in the Figure and what the abbreviations are.
Author Response
This review article by Fu et al. explores the intricate relationship between the gut microbiota and hepatic cancers, focusing on hepatocellular carcinoma (HCC) and cholangiocarcinoma (CCA). It synthesizes current literature on genetic and epigenetic alterations, microbiota-driven inflammatory and metabolic changes, and host–microbe interactions that influence hepatic tumorigenesis. The review also evaluates the diagnostic and therapeutic potential of microbiota modulation and outlines gaps in the field. The manuscript is comprehensive, well-organized, and addresses a timely and increasingly relevant research area, with strong potential appeal to both basic scientists and clinicians.
Specific Comments
- Line 348: The figure legend for Figure 1 is missing any explanatory detail beyond the title. A brief description of key pathways illustrated would enhance clarity.
Thank you very much for your comment. The figure legends have been rewritten in detail, and the abbreviations have been explained accordingly.
- Lines 427–458: The coverage of microbiota’s role in CCA is less comprehensive than for HCC. While liver fluke–associated CCA is well-described, further discussion of non-infectious dysbiosis in CCA would be helpful.
Thank you very much for your comment. We have rewritten the entire paragraph, adding evidence-based discussion of non-infectious dysbiosis in CCA.
- Lines 498–533: The enthusiasm about microbiota as a diagnostic/prognostic tool is appropriate and well-timed, but a brief mention of confounding factors, such as population, diet, sequencing platform would add necessary balance.
Thank you very much for your comment. We have rewritten the entire paragraph.
- Lines 503–533: The prognostic role of the microbiota is well discussed, but the section would benefit from clearer distinction between correlation and causation. This is especially important because these findings are derived from human cohort studies rather than preclinical/mechanistic studies. Consider flagging which data come from human cohorts vs. animal models.
Thank you very much for your comment. We have rewritten the entire paragraph.
- Line 563: The statement that FMT has potential applications in “other age-related diseases” is out of scope unless the authors briefly elaborate or remove this phrase.
Thank you very much for your comment. The part “and other aged-related diseases” has been deleted.
- Line 580–593: The section on immune checkpoint inhibitors focuses largely on melanoma studies. Consider briefly discussing what (if anything) is known in HCC/CCA models or clinical cohorts.
Thank you very much for your comment. We have rewritten the entire paragraph.
- Lines 594–641: Strong section overall. Consider citing specific examples where liver organoid-microbiota co-cultures have been used, if available.
This is a very interesting commentary. Unfortunately, liver organoids are very complex to perform and no data on co-culture with microbiota are available at the moment. Nevertheless, we thank the reviewer for this comment, as it has opened a portal for us to write a future project based on this idea.
- Line 671: Figure 2 legend needs to be expanded to describe everything depicted in the Figure and what the abbreviations are.
Thank you very much for your comment. The figure legends have been rewritten in detail, and the abbreviations have been explained accordingly.
